# Specific Features of Focal Cortical Dysplasia in Tuberous Sclerosis Complex

**Ekaterina Bychkova** [1,2,*], **Marina Dorofeeva** [3], **Aleksandr Levov** [4], **Alexey Kislyakov** [4], **Kristina Karandasheva** [1], **Vladimir Strelnikov** [1] and **Kirill Anoshkin** [1]

1. Research Centre for Medical Genetics, Moskvorechye Street 1, 115522 Moscow, Russia
2. Faculty of Biomedicine, Pirogov Russian National Research Medical University, Ostrovityanova Street 1, 117997 Moscow, Russia
3. Veltischev Research and Clinical Institute for Pediatrics and Pediatric Surgery, Pirogov Russian National Research Medical University, Taldomskaya 2, 125412 Moscow, Russia
4. Morozov Children's City Clinical Hospital, 4th Dobryninsky Lane, 1/9, 119049 Moscow, Russia
* Correspondence: ktrn.bychkova@gmail.com; Tel.: +7-(920)-923-27-73

**Abstract:** Patients with tuberous sclerosis complex present with cognitive, behavioral, and psychiatric impairments, such as intellectual disabilities, autism spectrum disorders, and drug-resistant epilepsy. It has been shown that these disorders are associated with the presence of cortical tubers. Tuberous sclerosis complex results from inactivating mutations in the *TSC1* or *TSC2* genes, resulting in hyperactivation of the mTOR signaling pathway, which regulates cell growth, proliferation, survival, and autophagy. *TSC1* and *TSC2* are classified as tumor suppressor genes and function according to Knudson's two-hit hypothesis, which requires both alleles to be damaged for tumor formation. However, a second-hit mutation is a rare event in cortical tubers. This suggests that the molecular mechanism of cortical tuber formation may be more complicated and requires further research. This review highlights the issues of molecular genetics and genotype–phenotype correlations, considers histopathological characteristics and the mechanism of morphogenesis of cortical tubers, and also presents data on the relationship between these formations and the development of neurological manifestations, as well as treatment options.

**Keywords:** tuberous sclerosis complex; tuber; *TSC1*; *TSC2*; mTORC1; epilepsy

## 1. Introduction

Tuberous sclerosis complex (TSC), also known as Bourneville–Pringle disease, is an autosomal-dominant disorder classified as phakomatosis, with an incidence of 1 per 6000–10,000 live births [1–3]. The products of the *TSC1* and *TSC2* genes, hamartin and tuberin, together with the TBC1D7 protein, form a complex that implements negative regulation of the mTOR signaling pathway, which is responsible for controlling cell growth and autophagy. Inactivating mutations in *TSC* genes provoke sustained activation of RHEB, a GTP-binding protein that activates the mechanistic target of rapamycin (mTOR). This, in turn, induces phosphorylation cascades, leading to cell growth, proliferation, and the suppression of autophagy [1,4].

The phenotypic manifestations of TSC are widely variable. These include heart, lung, kidney, skin, and brain malformations [1,3]. The absence of pathognomonic features complicates the diagnosis. Diagnosis can be made based on a combination of at least 2 major features or 1 major and 2 minor features. Major features comprise hypomelanotic macules (≥3, at least 5 mm in diameter), angiofibromas (≥3) or fibrous cephalic plaques, ungual fibromas (≥2), a shagreen patch, multiple retinal hamartomas, cortical dysplasias, subependymal nodules, subependymal giant cell astrocytoma, cardiac rhabdomyoma, lymphangioleiomyomatosis, and angiomyolipomas (≥2). Minor features comprise confetti skin lesions, dental enamel pits (>3), intraoral fibromas (≥2), a retinal achromic patch,

multiple renal cysts, and nonrenal hamartomas. The presence of a pathological mutation in the *TSC1* or *TSC2* genes is sufficient for a diagnosis of TSC [1].

The presence of cortical tubers, subependymal nodes (SENs), or subependymal giant cell astrocytomas (SEGAs) in the cerebral cortex and/or subcortical white matter is a distinctive histopathological characteristic of TSC in the brain that occurs in 80–90% of cases [2]. Epilepsy is present in 80–90% of TSC cases and usually develops before the age of three [1–3]. It has been shown that disorders such as intellectual disabilities, autism spectrum disorders, and drug-resistant epilepsy, which occur in patients with TSC [2,5], are associated with the presence of cortical tubers [2,6,7]. The mechanism of epilepsy is a complex process that involves more than one signaling pathway, so it remains underdetermined.

Tubers are malformations of cortical development, represented by a disruption of the normal hexalaminar structure of the cerebral cortex, the formation of atypical neurons and glial cells, and a significant decrease in the number of normal neurons [2,8]. Abnormal cells are represented by dysmorphic neurons (DNs), giant cells (GCs), and gliotic and reactive astrocytes [2,3,9,10]. Hypomyelination, an abnormal vascular density, and inflammation are detected in the tuber tissue [11–13]. Patients with TSC most frequently have multiple tubers of various sizes and shapes [5]. Cortical tubers are formed during the prenatal period and stay dynamic after birth [14]. Tubers are observed within each lobe of the cerebral cortex and superficial white matter and can also be found in the cerebellar cortex [5,15,16]. A higher tuber count is strongly associated with the presence of infantile spasms [17], and in about half of cases, tubers undergo cystic transformation or calcification, and these are associated with more severe epilepsy [11,18].

The *TSC1* and *TSC2* genes are classified as tumor suppressor genes and function according to Knudson's two-hit hypothesis, which means that both alleles must be damaged for tumor formation. However, a second-hit mutation is a rare event in cortical tubers [19,20]. This suggests that the molecular mechanism of cortical tuber formation may be more complicated and requires further research.

## 2. Structural Features of Cortical Tubers

Tubers are areas of cortical dysplasia represented by several types of atypical, enlarged cells, including DNs, gliotic and reactive astrocytes, and GCs [2,9,10]. The normal hexalaminar structure of the neocortex is damaged within tubers. Classic features of focal cortical dysplasia, such as blurred boundaries of gray and white matter and cortex thickening, can be observed by MRI [11]. Microscopic structural changes in the brain, such as small areas of cortical dislocation, microtubers (small clusters of GCs, dysplastic astrocytes, and heterotopic neurons), and single, isolated GCs, are found in close proximity to the tubers [16,21,22]. These changes may contribute to the epileptogenicity of perituberal tissue. Cortical tubers should be considered a subtype of focal cortical dysplasia (FCD) type IIb because of their histological similarities: disruption of cortical lamination and the presence of morphologically abnormal cell types, specifically DNs and balloon cells (BCs), referred to as GCs in the case of TSC [23]. Another similarity is the molecular basis, consisting of the hyperactivation of mTOR signaling [24]. The difference between TSC and other forms of FCD IIb lies in the concomitant clinical and imaging features required to fulfill the TSC diagnostic criteria [23].

MRI is widely recognized as the reference method for defining CNS involvement in TSC. In infants, tubers appear as localized cortical thickening with moderate hyperintensity on T1-weighted images and hypointensity on T2-weighted images compared with normal unmyelinated tissue. After myelin maturation, tubers typically appear as areas of increased signal from the cerebral cortex and decreased signal from the subcortical white matter on T1-weighted sequences, along with increased signals from the cortex and subcortical white matter on T2-weighted sequences and fluid-attenuated inversion recovery (FLAIR) images [15,25].

Based on their MRI appearance, tubers have been classified into different types, including isointense on volumetric T1 images and subtly hyperintense on T2-weighted and FLAIR

images (type A), hypointense on volumetric T1 images and homogeneously hyperintense on T2-weighted and FLAIR images (type B), and hypointense on volumetric T1 images, hyperintense on T2-weighted, and heterogeneous on FLAIR images characterized by a hypointense central region surrounded by a hyperintense rim (type C) [26]. Type C tubers correspond to cyst-like tubers, which are found in about 50% of TSC cases at an early age and are more common with *TSC2* mutations [14,18]. Cyst-like tubers are expected to result from cellular degeneration or apoptosis. Gliosis is stronger in cystic tubers and the perituberal region than in conventional tubers. Cystic tubers are characterized by a more significant loss of subcortical white matter than classical tubers. It has been shown that an increase in the size of cystic tubers and the formation of cystic cavities in conventional tubers can occur in the postnatal period [14]. The presence of cystic tubers strongly correlates with more severe epilepsy [18]. Calcified tubers occur in 54% of patients with the diagnosis of TSC and progress with age [27]. The presence of calcified tubers correlates with more severe seizures and an early onset of the disease [11]. In addition, it has been shown that calcification in epileptic foci indicates a drug-resistant type of epilepsy and may, therefore, serve as its marker [27].

In cortical tubers, the following types of atypical cells are represented: DNs, GCs, and dysplastic astrocytes. DNs have an enlarged soma, atypical processes, a high content of neurofilaments, and Nissl bodies in the cytoplasm [11,28]. These abnormal cells express neurofilament proteins in their cell bodies. DNs can be located within all cortical layers as well as in subcortical white matter [8,29]. They are immunoreactive to p-S6, a downstream mTOR substrate, demonstrating cell-specific activation of the mTOR pathway [16,29], and are positive for neuronal markers such as NF200, DCX, and NeuN [30]. DNs and GCs show nuclear and cytoplasmic accumulation of p62, a stress-inducible intracellular protein that is known to regulate different signal transduction pathways and has been identified as a key target of autophagy [31].

GCs detected in TSC are histologically similar to BCs of FCD type IIb [31]. These are enlarged cells with an eosinophilic opalescent cytoplasm and one or more flattened and decentered nuclei [30,31]. GCs have a cell body that is 3 to 10 times greater in length compared to common neurons and glial cells [9]. They are mainly present in the deep layers of the affected cortical area and corresponding white matter as well as outside TSC lesions [31]. An immunohistochemical and electron microscopy analysis indicated the presence of two different subtypes displaying glial or neuronal features [9,30]. Electron microscopy revealed neuronal features, such as round nuclei with a prominent nucleolus, an organized reticulum and microtubules, lots of intermediate filament bunches, and synapses on processes, and glial features, such as a large, smooth cytosol devoid of organelles around the nucleus, an accumulation of organelles at the periphery of the cell, and numerous glial filaments and lysosomes [9,30]. An immunohistochemical analysis showed that some GCs express markers of mature neurons (neurofilaments, NeuN, synaptophysin, doublecortin, MAP2, neuron-specific enolase, neuropeptide Y) and those of immature neurons (doublecortin). Other GCs are positive for markers of glial cells (GFAP, β-crystallin, S-100) [9]. Markers of neural stem cells (SOX2, nestin, CD133) are also expressed by GCs. Interestingly, some GCs have been reported to be immunopositive for interneuron markers (GABA, parvalbumin, calbindin, calretinin) [30]. GCs are immunopositive for p-S6, indicating hyperactivation of the mTOR pathway [30,31]. The expression of P-gp was also observed in GCs, DNs, and reactive astrocytes. This multidrug resistance phenotype may function as a general protective mechanism during early development; the overexpression of P-gp may diminish the response to antiepileptic drugs [28].

Astrocytes are presented in tubers in a significantly larger amount than in the perituberal cortex and control brain tissue [10,32]. Two subpopulations of abnormal astrocytes have been distinguished within tubers—gliotic and reactive. Gliotic astrocytes are characterized by smaller sizes and lower levels of glutamate transmitters, glutamine synthetase, and inward-rectifier potassium channels, all of which represent proepileptogenic changes [10]. Reactive astrocytes have larger soma and the presence of thick and tortuous extensions and

express GFAP and vimentin [10,30]. The expression of p-S6 in reactive astrocytes reveals mTOR activation, although compared with DNs and GCs, dysplastic astroglia represent only a smaller fraction of cells expressing the p-S6 protein [10,30]. Whether the different degrees of mTOR activation underlie the wide diversity in astrocyte functions and phenotypes in TSC is a major research challenge. The dynamics of the distribution of gliotic and reactive astrocytes may be responsible for the changes in epileptogenicity and the morphological dynamics of tubers over time [32]. Astrocytes demonstrate high levels of ICAM-1, NFkB, TNFa, and IL-1b and its signaling receptor IL-1RI, as well as greater expression of the enzymes inducible nitric oxide synthase (iNOS) and cyclooxygenase 2 (COX-2) [10,32,33]. The expression of proinflammatory cytokines and inflammatory enzymes may further activate astrocytes within tubers and foster epileptogenesis.

Subcortical white matter loss is also detected in tubers, which may be explained by a loss of projections between abnormal neurons and the other brain regions. Significant myelin reduction has been shown in tubers, but it is not always accompanied by a decrease in the number of oligodendrocytes [11,34]. Near the base of the cortical tuber, clusters of cytomegalic neurons commonly form heterotopias that are accompanied by hypomyelination [35]. Myelination defects are assumed to be a consequence of the impaired reciprocal regulation of neurons and oligodendrocytes [36].

As in SENs and SEGAs, inflammation is present in tubers, albeit to a lesser degree [20]. The expression of HLA-DR, CCL2, and SerpinA3 indicates the activation of microglia within tuber tissue [37]. Microglia are often clustered around DNs and GCs and are associated with immune mediators, such as CD8-positive T-lymphocytes and the complement cascade. Cortical tubers show a large number of activated microglial cells expressing MHC class II antigens [12]. Microglia activation has a damaging effect on oligodendrocytes and neurons, leading to neuronal dysfunction, concomitant neurological diseases, and increased excitability [32]. Proinflammatory factors expressed by microglial cells and the increased synthesis of vascular endothelial growth factor (VEGF) caused by mTOR hyperactivation are associated with changes in vascular density within tuber tissue. This increases the blood–brain barrier permeability in tubers compared to the surrounding perilesional cortex, promoting lymphocyte infiltration [13]. Zimmer et al. examined the transcriptional regulation of TSC tubers and found the expression of SPI1/PU.1 transcription factors in DNs and astrocytes with mTOR activation. SPI1/PU.1 is critical for the maturation and regulation of the immune system and brain inflammation and is a potential activator of toll-like receptor (TLR) signaling, complement activation, and cytokine signaling. In the healthy brain, SPI1/PU.1 is limited to microglia and infiltrating immune cells. It is suggested that targeting this transcription factor may reduce inflammation in TSC [38]. Higher expression of markers of oxidative stress (iNOS and xCT) and inflammation (TLR4 and COX-2) was detected in DNs and GCs/BCs and, to a lesser extent, in glia. In addition, the SH-SY5Y neuronal cell line was shown to respond to oxidative stress through the dose-dependent induction of inflammatory mediators, suggesting that oxidative stress is crucial for the development of the inflammatory process [33].

## 3. Molecular Genetics

### 3.1. Impaired Inhibition of the mTOR Pathway by the TSC Complex

The mTOR pathway is inhibited by a protein complex consisting of hamartin, tuberin, and TBC1D7. Hamartin is encoded by *TSC1* (chromosome 9q34.13), and tuberin is encoded by *TSC2* (chromosome 16p13.3). TSC1, TSC2, and TBC1D7 assemble the TSC complex with a 2:2:1 stoichiometry. Two TSC2 molecules come together to form a pseudo-symmetric dimer through tail-to-tail interactions, while the TSC1 dimer makes multiple contacts with the tuberin dimer and stabilizes the conformation of the complex. Dimerization of the TSC1 protein leads to asymmetric formation of the TSC1–TSC2 tetramer and the recruitment of a single TBC1D7 molecule [39]. Hamartin and tuberin stabilize each other, which is confirmed by a significant decrease in the TSC1 protein with a homozygous *TSC2* mutation and, vice versa, a decrease in the TSC2 protein with a homozygous *TSC1*

mutation [40,41]. It has been demonstrated that, as a result of inactivating mutations in the *TBC1D7*, the association between hamartin and tuberin is partially disrupted, but knockdown of the *TBC1D7* gene is not the cause of TSC [42]. However, homozygous loss of *TBC1D7* causes intellectual disability and megalencephaly, a developmental disorder associated with mTORC1 activation [43].

The TSC1–TSC2–TBC1D7 complex regulates the activity of the mTOR protein kinase by inhibiting the regulatory protein RHEB. Regulation of the mTOR signaling pathway is carried out by a GAP domain in tuberin, which hydrolyzes GTP bound to RHEB, thereby inactivating the protein [44]. TSC1 is believed to enhance TSC2 function by activating TSC2 GAP activity, stabilizing TSC2, and/or maintaining the correct intracellular localization of the TSC complex [43]. RHEB is an activator of the mTORC1 complex consisting of the mTOR protein kinase, the LST8 protein associated with the mTORC1 catalytic domain, RPTOR, a regulatory protein, and two inhibitory proteins, DEPTOR and AKT1S1 [7]. MTORC1 participates in the regulation of many cellular processes, including protein synthesis, lipogenesis, ribosomal biogenesis, purine and pyrimidine nucleotide biosynthesis, DNA transcription, autophagy, and mitochondrial biogenesis [44,45]. MTORC1 can be allosterically inhibited by rapamycin, a macrolide [4,44].

As a result of a *TSC1/TSC2* mutation, the GAP domain in TSC2 fails to perform its function, leading to impaired RHEB inhibition. The constitutive activity of RHEB leads to the hyperactivation of mTORC1. This leads to phosphorylation of the key effectors of protein synthesis, the p70S6 and 4E-BP1 kinases. The p70S6 kinase phosphorylates the ribosomal protein S6, resulting in the phosphorylation of transcription elongation factors, which facilitates translation and cell growth [4,44–46]. In addition, phosphorylation of the 70S6 kinase mediates lipid synthesis through SREBP activation [46]. The phosphorylation of 4E-BP1 stabilizes mRNA and increases the efficiency of initiation and elongation [44,46]. This results in increased translation, impaired cell migration, growth, and differentiation, and the suppression of autophagy (Figure 1). Dysregulation of this pathway during corticogenesis can dramatically complicate cortical lamination [46].

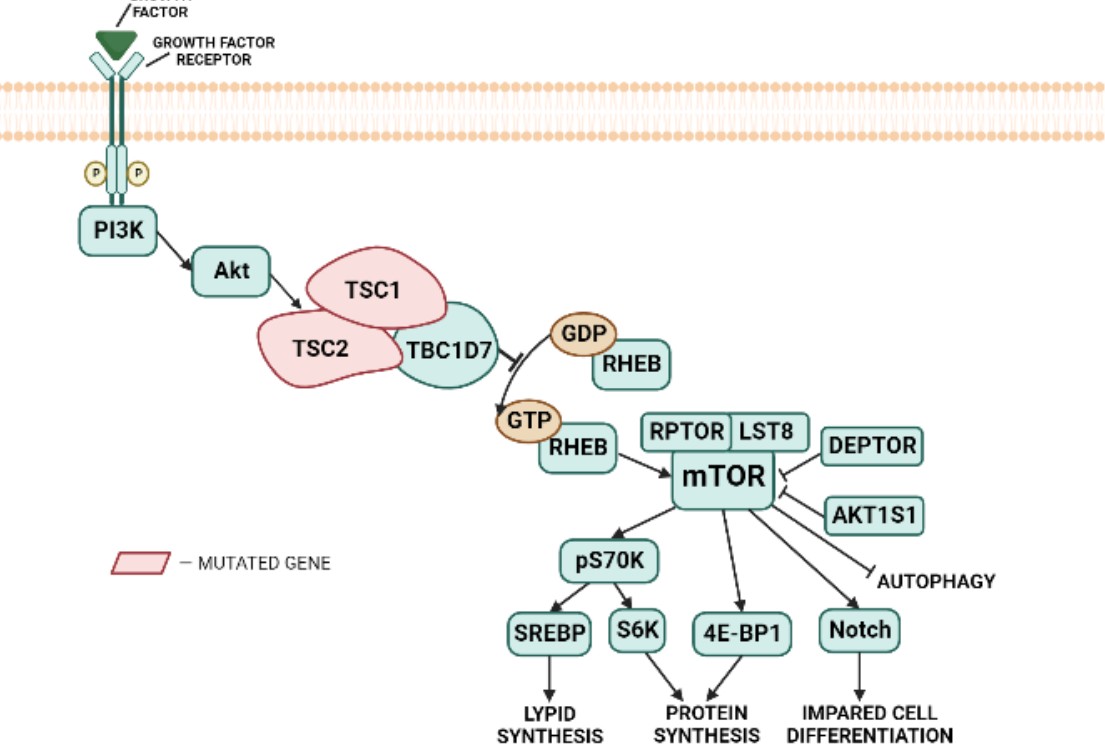

**Figure 1.** Effects of activation of the mTOR signaling pathway via TSC1 or TSC2 inactivation.

Studies have revealed that pathogenic *TSC1* and *TSC2* mutations are characterized by phenotypic manifestations of differing severity. Individuals with a *TSC1* mutation have a lower incidence of tubers, a lower average number of tubers, and a lower tuber load compared to patients with a *TSC2* mutation [47–49]. Several studies have confirmed that cyst-like tubers are more prevalent in patients with *TSC2* mutations [18,47]. Pathogenic mutations in the *TSC2* gene are also linked to an increased risk of developmental disorders, intellectual disability, and infantile spasms [47,50]. It was found that patients with *TSC2* pathogenic variants are significantly more likely to have developmental delays at the age of two years than patients with *TSC1* pathogenic variants or patients with no mutations identified [51]. Moreover, individuals with *TSC2* mutations tend to experience an earlier onset of infantile spasms (at around 1.2 years) than those with *TSC1* pathogenic variants (at around 2.7 years) [48,50,52]. Interestingly, no significant differences in mTORC1 activation and the severity of the mutations were found when comparing mutant mice with a mutation in *TSC1* to those with a *TSC2* mutation [53]. The second hit was made early in maturation in radial glia cells, so perhaps the phenotypic manifestations were too severe to spot the difference.

The disturbance of mTOR signaling plays a key role in a variety of neurological disorders, including TSC, FCD, hemimegalencephaly, epilepsy, and autism spectrum disorder [45,54]. A number of studies have identified somatic mutations in genes encoding canonical signaling proteins within the mTOR pathway, including *PIK3CA*, *AKT3*, *TSC1*, *TSC2*, and *MTOR* itself in the aforementioned disorders [54,55]. The mTOR signaling pathway plays vital roles in cortical development and the maintenance of cell homeostasis and energy metabolism and participates in the regulation of cell proliferation, differentiation, migration, and survival as well as in the formation and functioning of synapses [45,46]. This pathway is also involved in establishing the shape and size of neurons, dendritic arborization, spine morphology, axon outgrowth, and the regulation of excitatory and inhibitory neurotransmission [45]. MTORC1 was identified as a positive regulator of Notch signaling. By increasing Notch activation, newly formed neurons may prime multipotent progenitors to respond to gliogenic cytokines, resulting in a switch to astrogenesis [56,57]. The suppression of mTORC1 signaling during the transition from proliferation to neuronal differentiation of human neurons is a prerequisite for normal neuro- and gliogenesis [58]. One of the most important functions of mTORC1 in brain tissue is the regulation of the activity of mRNA translation near synapses, which plays a role in the mechanism of neural signal transmission [59]. Excessive activation of mTOR can also lead to impaired autophagy, disrupting the removal of abnormal proteins and damaged cell organelles [45,60]. Thus, dysregulation of this pathway during corticogenesis can complicate cortical lamination and lead to the formation of abnormally differentiated cells and pronounced cytoarchitectural disorganization. J. D. Blair et al. demonstrated the period (from days 12 to 80) of neurogenesis for pharmacological suppression of the mTOR pathway, which can prevent neuronal differentiation defects and cell hypertrophy in cortical spheroids. After this period, treatment with mTOR inhibitors may decrease mTORC1 hyperactivation and decrease neuronal and glial hypertrophy but may not fully restore normal cortical development [58].

The role of extracellular vesicles (EVs) has become an intriguing research issue. A significant increase in the production of EVs was demonstrated in *TSC*-null cell cultures as well as in cortical tubers [61]. It is believed that endoplasmic reticulum stress induced by *TSC* mutations and hypoxia from mTOR activation could be a potential trigger for EV production [62]. A gene expression analysis in cells incubated in culture with exosomes isolated from epileptogenic tissue revealed increased expression of genes related to the PI3K/AKT pathway. Dose-dependent increases in *SQSTM1* and *CDKN1A* were also shown. Both genes are linked to immune system signaling; *SQSTM1* is also reported to contribute to nutrient-sensing and the regulation of cellular metabolism through mTORC1. The higher expression of these genes may mediate epileptogenesis and exosome-induced activation on mTORC1 [61]. It is assumed that exosome release is regulated by mTOR in *TSC*-null cells under glucose deprivation, but exosome markers were increased even by treatment with

rapamycin under these conditions. Interestingly, exosome release inhibitors were found to attenuate rapamycin-treatment-induced cell viability in glucose-deprived *TSC*-null cells, suggesting that exosome release regulation provides a therapeutic way to control *TSC*-null cell survival [63]. Exosomes derived from epileptogenic tissue may be a prospective tool for the noninvasive monitoring of seizure susceptibility and the response to treatment, since they are stable, able to transit the blood-encephalic barrier, and circulate systemically [61].

### 3.2. Alternative Mechanisms for Brain Pathlogy in TSC

In addition to the pathogenetic role of mTOR hyperactivation, other mechanisms of tuber formation have been discussed. The alterations in cell signaling mechanisms critical for neuronal migration still remain unclear. One suggested mechanism of irregular migration is ineffective Reelin-Dab1 signaling. Reelin is a glycoprotein synthesized in the marginal zone during neurogenesis. It regulates neuronal migration by polarizing and stabilizing neuronal processes and ordering neuronal–glial connections. It functions by interacting with the VLDR and ApoER2 receptors, resulting in phosphorylation of their intracellular domain, Dab1. The phosphorylated Dab1 (pDab1) activates downstream pathways that regulate neuronal migration. The phosphorylation of Dab1 is regulated by the Src family kinase. Cullin 5 (Cul5), a core component of E3 ubiquitin-protein ligase complexes, is involved in the ubiquitin-mediated degradation of the pSrc kinase and pDab1. Cortical tubers from *TSC2* Emx1-Cre CKO mice show increases in Reelin, Dab1, and Cul5, but pDab1 levels are significantly reduced, resulting in the impaired regulation of migration [64]. Examination of the layer-specific neuronal markers SMI32, Tbr1, Satb2, Cux2, ER81, and RORβ in tuber tissue has revealed a general decrease in the number of neurons, as well as impaired cortical lamination, revealing neurons expressing layer-specific markers that are randomly scattered throughout the tuber volume and perituberal cortex [8,65]. Furthermore, the activation of caspase-3 and caspase-6 was detected in DNs, GCs, and astrocytes with more limited expression in oligodendrocytes, indicating apoptosis-mediated ongoing cell death within the dysplastic cortex [66].

It has been noted that PAK2 may be involved in the abnormal migration and generation of atypical cells. PAKs are a pleiotropic kinase family that is involved in various cellular physiological processes, especially neuronal migration. *TSC2*-null embryonic mouse fibroblasts show significant changes in the cytoskeleton, including the formation of numerous lamellipodium-like networks, increased disorganization of cytoskeletal actin, the appearance of prominent cortical actin structures, and the formation of many small microspikes, which can be partially eliminated by PAK2 inhibition. It has been demonstrated that the TSC1–TSC2 complex acts as a negative regulator of PAK2 and that the activation of PAK2 after the deletion of TSC1 or TSC2 occurs in an mTOR- and Rac-independent manner. RHEB is hypothesized to be a major regulator of PAK2 autophosphorylation activation [67].

### 3.3. Biallelic Mutations in the TSC1 or TSC2 Genes Are Rarely Detected in Cortical Tubers

Knudson's two-hit hypothesis has been shown to be valid for tumors in TSC, such as SEGA, renal angiomyolipoma, and facial angiofibroma [68–70]. According to this hypothesis, the inactivation of both alleles of a tumor suppressor gene is required for tumor formation. The first hit is an inherited or de novo germline mutation, and the second hit occurs in somatic cells, including a loss of heterozygosity or a point mutation, and leads to a complete loss of gene function [59,71]. Biallelic mutations in the *TSC1* or *TSC2* genes are rarely detected in tuber tissue samples, so the mechanism of the second hit in cortical tubers remains unclear [20,28,72]. Inactivation of the second allele of the tumor suppressor gene may be the result of epigenetic silencing; however, there are extremely few data on differential *TSC1* and *TSC2* methylation in cortical tubers to suggest that methylation may be responsible for the second hit [20,73]. There are also no mutations in tumor driver genes and mTOR pathway genes specific to TSC patients [20].

Given the rarity of detection of biallelic mutations in the *TSC1* or *TSC2* genes in tuber tissue samples, it is tempting to assume that the nature of *TSC* causative mutations makes Knudson's two-hit hypothesis nonapplicable to the pathogenesis of at least some TSC-associated lesions, such as cortical tubers. In 2020, Feliciano suggested that a loss of both functional copies of *TSC1* or *TSC2* is perhaps not necessary to cause TSC and that haploinsufficiency or dominant negative mutations could also underlie the pathogenesis of cortical tubers [5]. Mutations in the *TSC1* or *TSC2* genes that cause a premature termination codon (PTC) were demonstrated to be associated with nonsense mediated decay (NMD) of the RNA, resulting in a reduced level of the mutant transcript, thus presenting loss of function mutations [74,75]. In *TSC1*, virtually all mutations are nonsense or small frameshift changes producing PTCs [74], thus rendering them loss of function mutations. In *TSC2*, many different types of mutation are seen, including large deletions and rearrangements and a substantial proportion of missense and nonsense mutations. Theoretically, among missense mutations, those exerting dominant negative effects may be suspected. Hoogeveen-Westerveld et al. performed a functional assessment of dozens of *TSC1* and *TSC2* missense variants identified in individuals with TSC [40,41,76]. They demonstrated that, in most cases, the pathogenicity of *TSC* missense mutations is implemented through the instability of the mutated protein. No cases of increased protein stability associated with *TSC* missense mutations were found. This allows us to classify them as loss of function mutations, alongside PTCs produced by *TSC* nonsense and frameshift mutations.

About one-third of TSC cases are inherited in an autosomal dominant manner; the other cases are the result of de novo germline or somatic mutations [7,46,77]. Somatic mutations are reported to cause FCD, hemimegalencephaly, as well as TSC lesions. Somatic mutations occur in postzygotic cells during embryogenesis, and if a somatic mutation occurs late in development, it may be present in a percentage of cells in only one tissue, which makes such mutations difficult to identify [54]. Damaging somatic mutations need to be present in a threshold percentage of cells to disrupt neuronal development and function. It was reported that somatic mutations present in as few as 1% of cells can cause cortical malformation [78]. To detect somatic mutations present in a particular cell fraction, DNA from bulk tissue or single cells must be sequenced. One of the possible reasons for the absence of biallelic mutations from the brain tissue of TSC patients is that tubers include different cell types, and only a small portion of the tuber is affected by a second hit. Tubers contain atypical cells with an increased mTORC1 activity level as well as normal neurons and glial cells. In addition, tubers are infiltrated by immune cells that decrease the mutant allele frequency. Thus, the low allele frequency can make it difficult to identify a loss of heterozygosity in cortical tubers. P. B. Crino et al. performed DNA sequencing of individual P-S6 immunoreactive GCs isolated from the tissue of cortical tubers. Biallelic *TSC1* or *TSC2* mutations were found in these cells, while there were no second-hit mutations in the DNA isolated from peripheral blood and the DNA isolated from tuber tissue sections [79]. Mutations can also be located in noncoding regions, such as introns, promoters, and regulatory regions and, thus, cannot be detected using exome sequencing [58].

Human induced pluripotent stem cells (hiPSCs) and brain organoids have found wide applications in the study of cortical development [58,80]. In a study of neurons derived from hiPSCs, it was found that a monoallelic *TSC2* mutation was sufficient to cause a disturbance in the regulation of mTOR activity in neurons and led to enlarged neuronal soma and processes [80]. However, it was demonstrated that the altered expression of certain neuronal genes (*ITGB4*, *ITGA6*, *SRPX2*, *EZR*, *CD44*, and *CAPG*) identified in cortical tubers was observed only in neurons with biallelic *TSC2* mutations, while the expression of these genes in neurons with a monoallelic mutation remained unchanged, supporting the suggestion that a second hit only contributes to the severity of the symptoms [20]. It has been shown that these genes are involved in the regulation of cell adhesion and cell migration and encode proteins localized mainly in the extracel-

lular region, especially in exosomes. In addition, neurons with a homozygous mutation showed increased synchronicity of neuronal activity, which is one of the hallmarks of epilepsy. The hypersynchrony of neuronal activity was decreased in neurons treated with rapamycin, but these cells substantially increased their activity when the action of rapamycin was terminated [80]. Knudson's hypothesis is supported by a study of three-dimensional cortical spheroid development: a pattern of altered cortical development (decreased expression of neuronal markers and increased expression of glial markers in comparison with normal development) was observed only in cells with biallelic inactivation of the *TSC1* or *TSC2* genes, while spheroids with heterozygous mutations in *TSC1* or *TSC2* showed a normal profile of neuron and glia development [58]. Eichmüller et al. used human cerebral organoids from TSC-patient-derived iPSCs to determine whether biallelic inactivation is required for the initiation of tumor lesions. They hypothesized that subependymal lesions and cortical tubers develop from a heterozygous interneuron precursor, with the loss of heterozygosity arising at later stages during malformation development. Cells from tumor-like organoids remained heterozygous on days 135–160 of development and showed signs of mTOR activation at the same time, which suggests that a second hit at *TSC1* or *TSC2* is not required for tumor initiation, despite the appearance of a loss of heterozygosity during tumor progression. Interestingly, the GCs of tuber-like organoids maintained *TSC2* expression (98%) at day 230 of development, meaning that a second hit at *TSC2* is not required for tuber formation. In addition, interneuron precursors have low levels of *TSC1* and *TSC2* proteins, which may sensitize them to heterozygous mutations in *TSC* genes [81].

*3.4. Transcriptome and miRNA Expression in Cortical Tubers*

Studies of the protein-coding transcriptome revealed the altered expression of the spectrum of genes in cortical tubers and the perituberal region compared to normal brain tissue [20,82,83]. Using a single-cell RNA-Seq analysis, it was shown that most of the genes that were overexpressed were specific to microglia, and the genes that were underexpressed were specific to neurons [83].

Increased expression of genes involved in the regulation of innate and adaptive immunity, the inflammatory response, the cytokine-mediated signaling pathway, and cell adhesion has been shown (Table 1). The expression of immune and inflammatory molecules is believed to be increased by mTOR activation, since lower expression was observed in perituberal tissue. Elevated levels of cell adhesion and inflammatory molecules may contribute to increased permeability of the blood–brain barrier, the disorganization of cortical lamination, and epileptic activity in tubers [82].

Genes with reduced expression in cortical tubers are involved in the processes of neurogenesis, the regulation of glutamatergic and GABAergic synaptic transmission, as well as formation of inward-rectifier potassium channels (Table 1). Neural cell adhesion molecules (NCAM) contribute to the interaction between neurons during brain development, neuronal migration, synaptogenesis, and synaptic plasticity. Contactin-3, an immunoglobulin-like NCAM, is downregulated in cortical tubers at the RNA and protein levels, especially during the early postnatal period, which is critical for brain development. The lack of contactin-3 may be involved in the development of neurodevelopmental and behavioral disorders in TSC [84]. Alterations in glutamatergic and GABAergic synaptic transmission are critically involved in hyperexcitability and epileptogenesis as well as in the buffering of the extracellular potassium levels [31].

**Table 1.** Altered expression of genes involved in inflammation, cell adhesion, neurogenesis, the regulation of glutamatergic and GABAergic synaptic transmission, and the formation of inward-rectifier potassium channels in cortical tubers.

| Gene | Function of the Encoded Protein | Expression Level |
|---|---|---|
| CCL2 | Recruits monocytes, memory T-cells, and dendritic cells to the inflammation sites | Increased |
| CCL3 | Participates in acute inflammation, attraction, and activation of polymorphonuclear leukocytes | Increased |
| CCL4 | Chemoattractant for natural killer cells, monocytes, and other immune cells | Increased |
| SERPINA3 | Participates in the regulation of inflammation and the immune response | Increased |
| CX3CR1 | A receptor for the chemokine CX3CL1 involved in the adhesion and migration of lymphocytes | Increased |
| ECM2 | Participates in cell adhesion | Increased |
| VCAM1 | Participates in the adhesion of leukocytes and endothelial cells and signal transmission | Increased |
| CD44 | A hyaluronic acid receptor involved in the activation of lymphocytes and the recirculation and homing of hematopoietic cells | Increased |
| DCLK1 | Promotes neuronal migration during neurogenesis | Increased |
| LTF | Participates in iron ion binding and transport, has antibacterial, antiviral, antiparasitic, catalytic, anticancer, and antiallergic properties | Increased |
| GAD1 | Catalyzes the conversion of glutamate to GABA | Decreased |
| GLT1D1 | Promotes the elimination of glutamate from the synaptic gap | Decreased |
| GABRA5 | The GABA A receptor $\alpha5$ subunit | Decreased |
| KCNJ3 | Participates in the reduction of resting membrane potential during hyperpolarization by conducting a weak potassium current inside the cell | Decreased |
| NCAM1, NCAM2 | Transmit a signal that induces neurite growth, participate in cell adhesion | Decreased |

Several studies have shown changes in miRNA expression in cortical tuber tissue. Increased expression of the miR-34 family miRNAs (miR-34a, miR-34b, and miR-34c), which participate in corticogenesis and the signal transduction of glutamate receptors, has been found. The increased expression of miR-34b-5p can activate inflammation in reactive astrocytes [83]. Significant upregulation of the inflammation-related miRNAs miR21, miR146a, and miR155 has been demonstrated in DNs, GCs, and reactive astrocytes within the tuber. These miRNAs appear to contribute to the regulation of the activation of IL-1R/TLR signaling and the astrocyte-mediated inflammatory response [85]. Overexpression of miR-142-3p, miR142-5p, miR223-3p, miR32-5p, and miR200b-3p was shown in epileptic tubers. These miRNAs downregulate the expression of genes involved in various neurological disorders, including epilepsy (*SLC12A5*, *SYT1*, *GRIN2A*, *GRIN2B*, *KCNB1*, *SCN2A*, *MEF2C*, and *TSC1*) [86]. Increased levels of miR-142-3p, miR-223-3p, and miR-21-5p have been detected within epileptogenic extracellular vesicles (EVs), which has been confirmed to activate TLR7/8 [61]. Increased mTOR activation is expected to lead to the induction of miR-23a and miR-34a, mediated by the activation of the p53 protein. These microRNAs reduce the expression of genes involved in synaptic transmission and neuronal development, which plays a role in the formation of epileptic tubers [87]. Inflammation in epileptic tubers, in turn, provokes additional abnormal expression of microRNA [85,87].

## 4. The Origin of Cortical Tubers

### 4.1. Normal Cortical Development and Tuberogenesis

The human cerebral cortex develops from the neuroectoderm. Neuroepithelial cells (NECs) divide symmetrically, increasing their population pool. During neural tube formation, NECs transform into radial glia cells, and together, line the lateral ventricles, thus forming the ventricular zone [88]. These radial glia cells give rise to different types of brain cells, including neurons, astrocytes, oligodendrocytes, and ependyma, in a temporally regulated manner [5]. It is assumed that the population of radial glia cells is heterogeneous, comprising cells committed to neurons or macroglia as well as multipotent cells. About 10–20% of radial glia cells generate pairs of neurons. Radial glia cells also generate intermediate progenitor cells, which divide 1–3 times for self-renewal and generate pairs of neurons. Excitatory projection neurons migrate and constitute the cortical plate. Neurons are born in an inside-out order: at the early stage, neurons of the lower cortical layers are formed, followed by upper-layer neurons. Radial glia cells also provide a physical substrate for neuronal movement into the cortical layers. Finally, towards the end of neurogenesis, progenitors generate glial cells [88]. Correct cortical lamination is a step-by-step process, and each migration stage depends on the success of the previous one.

Tubers are thought to originate during fetal development, starting at around the 19th week of gestation (GW) [22,30,51–54,89,90]. The formation of dysmorphic astrocytes and GCs in subcortical zones are probably initial events in tuberogenesis. Dysmorphic astrocytes have been detected in abortive material at 19–21 GW, mainly in the marginal zone and subpial granule cell layer. GCs can be detected in the deep white matter by 23–24 GW, and the expression of neuronal markers in the GCs can be observed after only 30 GW. Clusters of GCs appear in the cortical plate by 32–34 GW [22,30]. It should be noted that the immunophenotypes of GCs in SEGA and tubers have been shown to be similar, which supports the hypothesis that these cells share a functionally related neuroglial progenitor cell [22]. The presence of DNs in the superficial and deep layers of the cortical plate is one of the latest events in tuber development: they can be observed at 36–39 GW. The presence of well-defined cortical tubers coincides with the appearance of DNs and bundles of neurofilaments at the bottom of the tuber [22,30].

The cell-of-origin for brain malformations in TSC remains unknown. To investigate the developmental processes unique to humans that are responsible for cortical malformations, Eichmüller et al. studied human cerebral organoids from iPSCs heterozygous for the pathogenic *TSC2* variant. They demonstrated caudal late interneuron progenitor (CLIP) cells as being common cells-of-origin for the neurodevelopmental disorder in TSC. Early lesions were found to consist almost exclusively of CLIP cell lineages, whereas other cell types appeared during disease progression [81].

### 4.2. Models for Studying the Development of Cortical Tubers

To better understand the formation of cortical malformations, researchers utilize mouse models. Although cortical tubers are not present in mice, they do exhibit similar abnormalities, and conditional *TSC1* and *TSC2* mutations are used due to the lethality of biallelic germline mutations at the embryonic stage [91]. Conditional mutations are performed at different stages of brain development, which allows for speculation about the timing of the second mutational event.

The loss of both copies of *TSC1* in dorsal NPCs of Emx1-Cre conditional knockout (CKO) mice resulted in a severe disruption of cortical lamination, an increased cell size, increased mTORC1 activity, astrogliosis, and hypomyelination. Astrocytes showed abnormalities in glutamate transport and potassium homeostasis, which plays a role in the development of seizure activity. It should be noted that heterozygous *TSC1* models had no seizures or increased mTORC1 activation [91]. Rapamycin treatment suppressed seizure activity, myelination, and astrocyte defects and led to a reduction in the soma size of hypertrophic neurons [91,92]. Notably, Emx1-Cre mice with *TSC1*-null neural progenitor

cells showed cortical thickening at 16.5 embryonic days, in line with the embryonic origin of cortical alterations in TSC patients [93].

*TSC2* hGFAP-Cre CKO mice, with a conditional mutation in radial glial cells, exhibited defects of cortical lamination, postnatal cortex thickening, and macrocephaly, enlarged cells, astrogliosis, and hypomyelination. An increase in the number of progenitor cells and a decrease in the number of postmitotic neurons, especially early neurons, was observed, suggesting impaired progenitor cell differentiation [93].

The SynI-Cre *TSC1* neuron-specific mouse model showed enlarged DNs with mTOR activation, impaired stratification, spontaneous seizure activity, and the presence of ectopic cells in the white matter and hippocampus. Seizure activity indicates the direct involvement of *TSC1*-deficient neurons. Enhanced glutamatergic neurotransmission was also shown. It is worth noting that the presence of severe hypomyelination can occur, even in the absence of oligodendrocyte defects [36]. Interestingly, no multinucleated GCs were found in the neuron-specific model, although they were present in Nestin-Cre *TSC1*-null models [94]. This is probably connected with the later implementation of the second hit after the onset of neuronal differentiation.

GFAP1-Cre mouse models, with preferential inactivation of *TSC1/2* in glial cells, but also in the subsets of neurons, especially in the hippocampus and cerebellum, showed a high degree of glial proliferation, neuronal disorganization, and consequently, megencephaly. It is important to note that the *TSC1* GFAP1 CKO model showed a milder phenotype compared with *TSC2* GFAP1 CKO. Interestingly, *TSC1* GFAP CKO and *TSC2* GFAP CKO mice had no significant differences in Glt1 expression, but the *TSC2* GFAP CKO had an earlier onset of epilepsy, a higher seizure frequency, and earlier mortality than *TSC1* GFAP1 CKO mice. This indicates that the severity of neurological disorders is determined by a combination of factors [95].

## 5. Epileptogenicity of Cortical Tubers and Its Treatment

Epilepsy is the most common symptom, affecting 85–90% of patients diagnosed with TSC [1,7]. Epilepsy in TSC is quite heterogeneous, frequently involving a variety of seizure types, including partial seizures, generalized seizures, and infantile spasms. TSC-associated epilepsy is characterized by early-onset seizures, starting in the first year of life in 63% of patients, often by three months of age, but the first seizure can also occur in adolescence or early adulthood [7,46]. Refractory to standard antiepileptic drugs, epilepsy develops in two-thirds of individuals with TSC [1]. Surgical removal of tubers causes the reduction or elimination of seizures, which supports tubers' contribution to seizure development [7]. It was observed that the tuber volume correlates with the severity of epilepsy and intellectual disability. The ratio of the tuber volume to the whole brain volume is supposed to be a more reliable indicator for the prediction of cognitive impairment [48].

EEG studies have made it possible to identify the seizure initiation zone. Epileptiform discharges appear to arise in the center of the epileptogenic tuber and spread to its periphery and to the perituberal cortex [96]. Some studies claim that the perituberal cortex has its own epileptogenic activity [1,96]. Tubers can be classified into three groups: drug-resistant epileptogenic tubers, drug-controlled epileptogenic tubers, and epileptogenically inert. All three types can be observed within the brain of a single patient. It is noteworthy that seizure-generating tubers and nonepileptogenic tubers can be indistinguishable morphologically [97]. Apart from EEG, different neuroimaging approaches have been used to identify epileptogenic tubers. PET with $\alpha$-[11C]-methyl-l-tryptophan (AMT), which traces the tryptophan metabolism via serotonin and/or kynurenine pathways, has been found to be effective for determining the epileptic focus by showing increased AMT uptake in the epileptogenic tubers. Moreover, AMT PET can identify a nonresected epileptogenic cortex in patients with a previously failed epilepsy surgery and can assist in planning reoperation [98]. PET with $^{18}$fluoro-2-deoxyglucose (FDG) determines the region with epileptic activity by reduced glucose metabolism. FDG-PET and MRI coregistration enables

more precise and noninvasive determination of the focus [99]. Single photon emission computed tomography (SPECT) using the technetium isotope compound Tc-99m determines regional cerebral blood flow, which is seen as secondary marker of neuronal activity. Ictal SPECT requires video–EEG; hence, it is applied in cases with discordant or negative MRI results [100].

The mechanism of tuber epileptic activity remains a topical issue of study, because it is essential for the development of effective treatment for patients with drug-resistant epilepsy (Figure 2). Hyperactivation of mTOR is directly related to epileptogenesis, which has been confirmed by the results of clinical studies (EXIST-3) [101] and the reduced number of epileptic seizures or their elimination following the use of mTOR inhibitors [7]. Persistent upregulation of the mTOR pathway is hypothesized to initiate changes in neuron physiology, dendritic morphology, and changes in dendritic spine density and structure that contribute to epileptogenesis [102]. Disorders of synaptic transmission of an excitatory neurotransmitter glutamate play a significant role in the mechanism of epileptogenesis. A critical factor that regulates neuronal excitability is the expression and function of synaptic receptors for the main excitatory neurotransmitter glutamate. Dysplastic cells have demonstrated different expression levels of ionotropic glutamate receptor subunits and decreased expression of glial glutamate transporters (EAAT1 and EAAT2) and glutamine synthetase, which may substantially contribute to increased network excitability in cortical tubers [10,29]. The role of glutamate transporter deficiency in astrocytes in epileptogenesis was shown in *TSC1* GFAP CKO mice. The injection of beta-lactam antibiotic ceftriaxone, which enhances glutamate transporter expression in astrocytes at an early age, reduces excitotoxic neuronal death and epilepsy severity in the TSC mouse model but does not completely eliminate seizure activity [103]. Interestingly, *TSC1* GFAP CKO and *TSC2* GFAP CKO mice have no significant difference in GLT1 expression, but *TSC2* GFAP CKO mice show an earlier onset of epilepsy, a higher seizure frequency, and earlier mortality compared with *TSC1* GFAP1 CKO mice. This indicates that the severity of neurological disorders is determined by a combination of factors [95]. Nestin-expressing GCs express receptor subunit profiles for the excitatory neurotransmitter glutamate, such as NR2D and GluR4, which are commonly seen in non-neuronal cortical cells and have been shown to be highly expressed in human cortical tuber GCs [94]. In addition, immature GABA receptors that are not able to take part in the transmission of inhibitory signals have been found in cytomegalic tuber cells. Rapid and sustained recovery from infantile spasms when taking vigabatrin, an inhibitor of GABA-degrading gamma transaminase, suggests that the GABAergic system is directly involved in the pathogenesis of infantile spasms [104]. These factors prevent the suppression of excitatory signals, leading to the hyperexcitation of cells, and can serve as a potential therapeutic target in epilepsy treatment. Altered expression of EAAC1, GluR, NR, and GABAAR mRNAs in tubers supports the hypothesis that seizures can be generated in tubers as a consequence of enhanced excitability [104].

Lower expression of the Kir 4.1 potassium channels is involved in the development of epilepsy by increasing extracellular potassium levels and neuronal hyperexcitability [10]. The role of microglial activation in the pathogenesis of epilepsy is controversial: it is not known whether the activation of microglia is a consequence of epileptic seizures or whether mTOR hyperactivation is the cause of microglial activation [105].

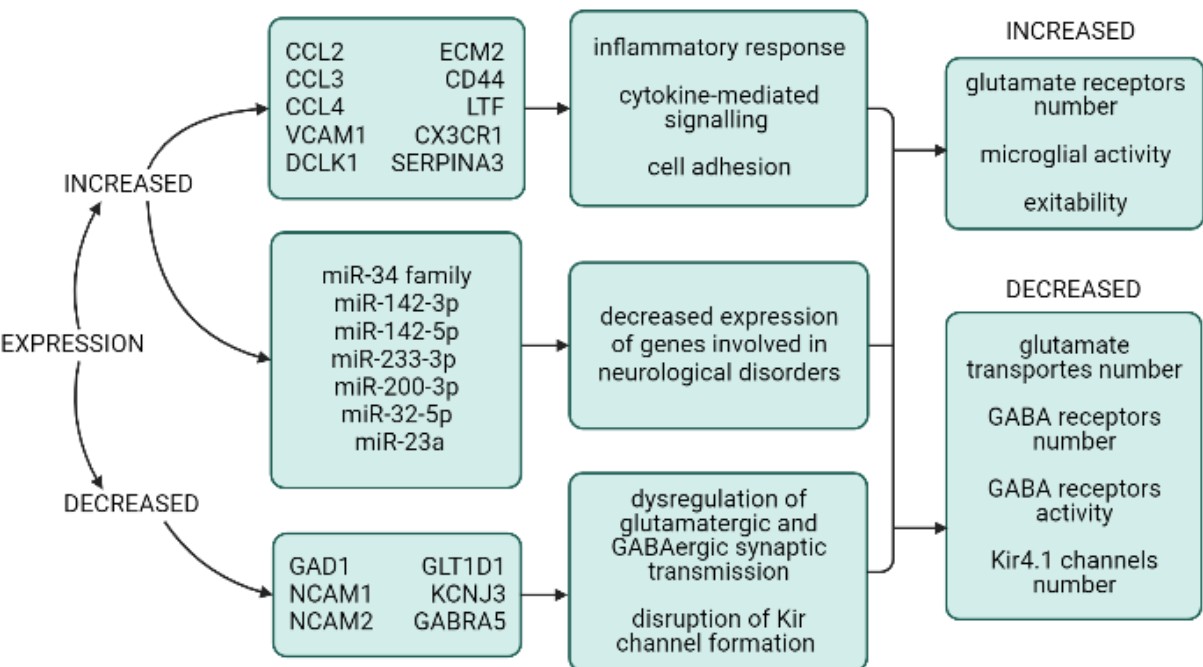

**Figure 2.** Possible factors of epileptic activity in cortical tubers in TSC.

The traditional approach to epilepsy treatment in TSC implies the initiation of treatment from the moment of epilepsy onset. However, epileptiform activity is detected on the EEG, even before the onset of clinical manifestations. In this regard, it is recommended that regular video–EEG monitoring of infants with the diagnosis of TSC should be conducted and therapy should be prescribed after the appearance of the first seizure focus on the EEG [106–108]. Drugs that control the action and content of GABA are mainly used as antiepileptic drugs (vigabatrin, clobazam) [7,46]. mTOR inhibitors, such as rapamycin or everolimus, have been shown to be effective against various types of tumors in TSC, but their efficiency against neurological symptoms is more limited. Rapamycin acts as an allosteric inhibitor that binds to and inhibits mTORC1 signaling. Long-term treatment with rapamycin can also prevent mTORC2 assembly, so the drug needs further modifications [109]. The rapamycin analogue everolimus demonstrated a marked antiepileptic effect in patients with drug-resistant epilepsy in the placebo-controlled EXIST-3 study [101,107].

One of the methods for drug-resistant epilepsy treatment is the resection of epileptogenic tubers in the brain [110]. Tuber resection spares patients from seizures in 50–60% of cases. In addition, removal of the perituberal cortex together with the epileptogenic tuber shows better results than tuber resection only [96]. Often, surgery does not bring complete relief from epilepsy for a number of reasons, such as uncertain localization of the epileptogenic tuber and failure to identify the whole focus. The goal of preoperative evaluation of a child with TSC is, first, to delineate the epileptogenic zone, and second, to determine the functional status of the cortex in and around the epileptogenic zone to assess the safety and risks of resection. Resection of the tubers alone may not completely eliminate seizure activity, because microscopic structural abnormalities are found in the vicinity of the tubers [16]. It is possible that these masses are responsible for the appearance of signs of epilepsy after tuber resection. Epilepsy surgery is an extremely complex operation that requires a carefully individualized approach. Several approaches have been developed, but none of them guarantee complete relief from seizures [7]. Further research is needed to develop the most successful strategies for the treatment of drug-resistant epilepsy.

## 6. Conclusions

Despite the abundance of available information about TSC, many aspects require further research. This review covers genetic aspects of TSC, the role of the mTOR signaling pathway in TSC pathogenesis, and the origin of cortical tubers, their histopathology, and their association with neurological diseases. It has been revealed that tuber morphogenesis begins during embryonic development. The main patterns of the cells present in cortical tubers have been discovered, and the connection between tubers and neuropsychiatric disorders, such as mental retardation, autism spectrum disorders, and epilepsy, has been proved. However, the exact causes of cortical dyslamination, the formation of abnormal cells, and the mechanism of epileptogenesis remain unrevealed. Studies of the two-hit mechanism of mutations in cortical tubers are contradictory, and the molecular genetic basis of their development is still unclear. It is likely that the timing and level of the second hit determine the extent of the lesions and their locations. This may determine the occurrence and severity of seizures and contribute to progressive cognitive impairment.

Due to the significant variation in TSC manifestations and their severity, it is necessary to study potential risk factors for specific symptoms. Over the past 10 years, significant results have been achieved in the treatment of the neurological features of TSC using an mTOR inhibitor, everolimus, but these drugs have a range of serious side effects and need further modifications. It is possible that Notch inhibitors, in combination with mTOR inhibitors, could produce a positive therapeutic effect. Further studies on the mechanism of cortical malformations and seizure activity in mTOR hyperactivation may reveal new therapeutic downstream targets to prevent the negative effects of mTOR inhibition. The resection of epileptogenic cortical tubers assisted with modern diagnostic methods has been successfully carried out in order to cure drug-resistant epilepsy. Nowadays, there are new approaches to epilepsy surgery, but this treatment still cannot provide total relief from seizures. Although the resection of cortical tubers is widely used in the treatment of epilepsy, the obtained material is often insufficient to conduct high-quality, reliable studies.

**Author Contributions:** Conceptualization, E.B. and K.A.; literature review and writing—original draft preparation, E.B.; writing—review and editing: M.D., A.L. and A.K.; data analysis, K.K.; visualization, E.B.; project administration, K.A. and V.S. All authors have read and agreed to the published version of the manuscript.

**Funding:** The research was carried out within the state assignment of Ministry of Science and Higher Education of the Russian Federation for RCMG.

**Institutional Review Board Statement:** Not applicable.

**Informed Consent Statement:** Not applicable.

**Data Availability Statement:** Not applicable.

**Acknowledgments:** The authors would like to thank Pavel Zhelnov for his help with the literature editing of this article.

**Conflicts of Interest:** The authors declare no conflict of interest.

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
