# Peer review of "Specific Features of Focal Cortical Dysplasia in Tuberous Sclerosis Complex"

_cimb, doi:10.3390/cimb45050254_

Round 1

Reviewer 1 Report

This article by Bychkova et al reviews the clinical, pathologic and molecular aspects of cortical tubers, cerebral lesions characteristic of focal cortical dysplasia in tuberous sclerosis complex (TSC).

This review is comprehensive, informative and well written, however, requires further editing:

1.    The use of disease name is inconsistent. “Tuberous sclerosis complex (TSC)”, rather than “tuberous sclerosis”, should be used in the title and abstract.

2.    There are several incomplete sentences and awkward expressions: eosinophilic opalescent [cytoplasm?] (line 127); expression of SPI1/PU.1 transcription factors [by?] DNs (lines 188-189); cell type caudal late interneuron progenitor [Remove “cell type”!] (lines 455-456); affected the reduction of [-> reduced?] (line 482); detected even before the onset of clinical manifestations on the EEG [-> detected on the EEG even before the onset of clinical manifestations?] (lines 613-614).

3.    There are several typographic errors: GCss [Remove one “s”!] (lines 77 and 464); ICAM-1, NFkB, TNFa[,] IL-1b (line 161); TSC2 (Do not italicize!) (line 228); 3.1. [-> 3.3.] (line 390); Figure 4 [Figure 3] (line 573).

Author Response

Dear reviewer,

Thank you for providing a precise analysis of the article and giving us the opportunity to submit a revised draft of our manuscript titled “Specific features of focal cortical dysplasia in tuberous sclerosis complex” to the Current Issues in Molecular Biology. We have been able to incorporate changes to reflect most of the suggestions provided by the review:

  1. Thank you for bringing this to our attention. We have taken your comment into account and corrected "Tuberous Sclerosis" to "Tuberous Sclerosis Complex" in the abstract and title.
  2. Thank you for your feedback. We have corrected all the listed mistakes, and we have also reviewed the text and corrected some grammatical and stylistic errors.
  3. Thank you for pointing this out. We have corrected all typographical errors. 

We also attempted to make the text more structured and logical, so we divided Section 4, "The Origin of Cortical Tubers", into two subsections and moved part of this section to subsection 3.2, "Alternative Mechanisms for Brain Pathology in TSC".

We look forward to hearing from you regarding our submission and to respond to any further questions and comments you may have.

Reviewer 2 Report

this a comprehenisve review on TSC and focal cortical dysplasia as it connects pathopysiology, diagnostic and genetics.

Author Response

Dear reviewer,

Thank you for providing your valuable feedback on our manuscript and giving us the opportunity to submit a revised draft of our manuscript titled “Specific features of focal cortical dysplasia in tuberous sclerosis complex” to the Current Issues in Molecular Biology. 

We reviewed the text of the article more carefully and corrected some grammatical and stylistic errors. We also attempted to make the text more structured and logical, so we divided Section 4, "The Origin of Cortical Tubers," into two subsections and moved part of this section to subsection 3.2, "Alternative Mechanisms for Brain Pathology in TSC". 

We look forward to hearing from you regarding our submission and to respond to any further questions and comments you may have.

Reviewer 3 Report

This paper is a punctual review on Focal cortical dysplasia inTuberous sclerosis.The topic is divided into 5 chapters according whit their specificity . While the 1 to 3 sections reported, are easy to follow, as well as the 5; for the section 4:' The origin of cortical tubers' I found it difficult to follow the progression of the exposition. The report is a frequent mix up of experimental and clinical data. The human versus the animal data have infact a different weight in  the interpretation of their role in the touberous origin. In my opinion the section 4 have to be revised and subdivided in human and experimental  data whit a syntesis of the current and more shared interpretation.

NB the number of the figure cited in the 573 line is 3, not 4 

Author Response

Dear reviewer,

Thank you for providing a precise analysis of the article and giving us the opportunity to submit a revised draft of our manuscript titled “Specific features of focal cortical dysplasia in tuberous sclerosis complex” to the Current Issues in Molecular Biology. We have been able to incorporate changes to reflect most of the suggestions provided by the review:

  1. Thank you for your comment regarding section 4 "The origin of cortical tubers". We tried to make the text more structured and logical, so we divided Section 4, "The Origin of Cortical Tubers," into two subsections and moved part of this section to subsection 3.2, "Alternative Mechanisms for Brain Pathology in TSC". 
  2. We reviewed the text of the article more carefully and corrected some grammatical and stylistic errors.

We look forward to hearing from you regarding our submission and to respond to any further questions and comments you may have.

Round 2

Reviewer 3 Report

The new version is satisfying. The sentence in 554/555 is not complete.

Author Response

Dear editor,

Thank you for reviewing our article again. We changed the sentence to make it complete. It is on line 531 in the new version of the article.